# Part-Based 3D Face Morphable Model
# with Anthropometric Local Control

Donya Ghafourzadeh*
Ubisoft La Forge, Montreal, Canada

Cyrus Rahgoshay†
Ubisoft La Forge, Montreal, Canada

Sahel Fallahdoust‡
Ubisoft La Forge, Montreal, Canada

Andre Beauchamp§
Ubisoft La Forge, Montreal, Canada

Adeline Aubame¶
Ubisoft La Forge, Montreal, Canada

Tiberiu Popa‖
Concordia University, Montreal, Canada

Eric Paquette**
École de technologie supérieure, Montreal, Canada

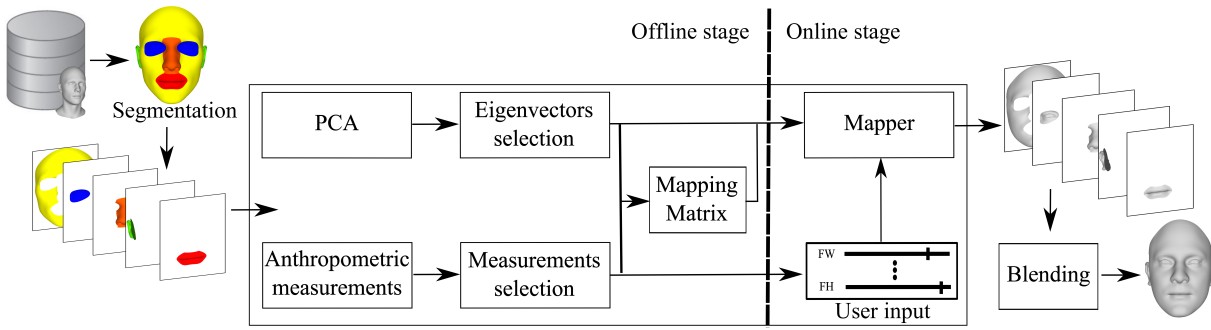

Figure 1: Our 3D facial morphable model workflow. In an offline stage, we extract PCA eigenvectors and select the best ones. We also select the best subset of anthropometric measurements. The relationship between the eigenvectors and measurements is encoded in a mapping matrix. All of these are used in the online stage, where the mapper collects user-prescribed anthropometric measurement values, and applies the mapping matrices to reconstruct the parts. The last step provides the edited face through a smooth blending of the parts.

## ABSTRACT

We propose an approach to construct realistic 3D facial morphable models (3DMM) that allows an intuitive facial attribute editing workflow. Current face modeling methods using 3DMM suffer from a lack of local control. We thus create a 3DMM by combining local part-based 3DMM for the eyes, nose, mouth, ears, and facial mask regions. Our local PCA-based approach uses a novel method to select the best eigenvectors from the local 3DMM to ensure that the combined 3DMM is expressive, while allowing accurate reconstruction. The editing controls we provide to the user are intuitive as they are extracted from anthropometric measurements found in the literature. Out of a large set of possible anthropometric measurements, we filter those that have meaningful generative power given the face data set. We bind the measurements to the part-based 3DMM through mapping matrices derived from our data set of facial scans. Our part-based 3DMM is compact, yet accurate, and compared to other 3DMM methods, it provides a new trade-off between local and global control. We tested our approach on a data set of 135 scans used to derive the 3DMM, plus 19 scans that served for validation. The results show that our part-based 3DMM approach has excellent generative properties and allows the user intuitive local control.

*e-mail: donya.ghafourzadeh@ubisoft.com
†e-mail: cyrus.rahgoshay@ubisoft.com
‡e-mail: sahel.fallahdoust@ubisoft.com
§e-mail: andre.beauchamp@ubisoft.com
¶e-mail: adeline.aubame@ubisoft.com
‖e-mail: tiberiu.popa@concordia.ca
**e-mail: eric.paquette@etsmtl.ca

**Index Terms:** Computing methodologies—Computer graphics—Shape modeling—Mesh models

## 1 INTRODUCTION

The authoring of realistic 3D faces with intuitive controls is used in a broad range of computer graphics applications, such as video games, person identification, facial plastic surgery, and virtual reality. This process is particularly time-consuming, given the intricate details found in the eyes, nose, mouth, and ears. Consequently, it would be convenient to use high-level controls, such as anthropometric measurements, to edit human-like character heads.

Many methods use 3D morphable face models (3DMM) for animation (blend shapes), face capture, and face editing. Even though face animation concerns are important, our work focuses on the editing of facial meshes. 3DMMs are typically constructed by computing a Principal Component Analysis (PCA) on a data set of scans sharing the same mesh topology. New 3D faces are generated by changing the relative weights of the individual eigenvectors. These methods are popular due to the simplicity and efficiency of the approach, but suffer from two fundamental limitations: they impose global control on the new generated meshes, making it impossible to edit a localized region of the face, and the control mechanism is very unintuitive. Some methods compute localized 3DMMs but those focus on facial animation instead of face modeling. We compared our approach to previous works relying on facial animation and saw that their automatic localized basis construction works well for animation purposes (considering a data set composed of animations for a single person), but performs worse than our approach for modeling purposes (considering a data set made of neutral faces from different persons).

We propose an approach to construct realistic 3DMMs. We increase the controllability of our faces by segmenting them into independent sub-regions and selecting the most dominant eigenvectors per part. Furthermore, we rely on facial anthropometric

measurements to derive useful controls to use in our 3DMM for editing faces. We propose a measurement selection technique to bind the essential measurements to the 3DMM eigenvectors. Our approach allows the user to edit faces by adjusting the facial parts using sliders controlling the values of anthropometric measurements. The measurements are mapped to eigenvector weights, allowing us to compute the individual parts matching the values selected by the user. Finally, the reconstructed parts are seamlessly blended together to generate the desired 3D face.

## 2 RELATED WORK

3D morphable models are powerful statistical models widely used in many applications in Computer Vision and Computer Graphics. One of the most well-known previous works in this regard is that by Blanz and Vetter [3]. Their pioneer work proposes a model using PCA from face scans. Although they propose a multi-segment model and decompose a face into four parts to augment expressiveness, the PCA decomposition is computed globally on the whole face. Other global PCA methods have been proposed [1, 4, 5, 8, 17, 18]. A downside of global PCA-based methods is that they exhibit global support: when we adjust the eye, the nose may also undergo undesirable changes. Another downside is a lack of intuitive user control for face editing. While the eigenvectors are good at extracting the dominant modes of variation of the data, they provide weak intuitive interpretation.

To address the former problem, local models have been proposed. They segment the face into independent sub-regions and select the most dominant eigenvectors per part. Tena et al. [26] propose a method to create localized clustered PCA models for animation. They select the location of the basis using spectral clustering on the geodesic distance and a correlation of vertex displacement considering variations in the expressions. Their method requires a manual step to adjust the boundaries of the segments, making it somewhat similar to ours, where the parts are user-specified. Chi et al. [9] adaptively segment the face model into soft regions based on user-interaction and coherency coefficients. Afterwards, they estimate the blending weights which satisfy the user constraints, as well as the spatio-temporal properties of the face set. Here too, the required user intervention renders the segmentation somewhat similar to our user-provided segments. SPLOCS [19] propose the theory of sparse matrix decompositions to produce localized deformation from an animated mesh sequence. They use vertex displacements in the Euclidean coordinates to select the basis in a greedy fashion. We noticed that when considering variation in identity instead of variation in expression, the greedy selection leads to bases which are far less local than those obtained from both our method and Tena et al.'s [26]. These papers address facial animation instead of face modeling and therefore assume large, yet localized deformations caused by facial expressions, which are different from our context where each face is globally significantly different from the others.

Like Tena et al. [26], Cao et al. [7] segment the face with the same spectral clustering, followed by manual adjustment. While their method focuses mostly on expression, they also provide some identity modeling, as they rely on the FaceWarehouse [8] global model, which they decompose using the segments defined by spectral clustering. In their case, the goal is to adapt a 3DMM to a face from a video feed, in real time. While their method works remarkably well for the real-time "virtual makeup" application, it lags behind ours in terms of providing a very detailed facial model, and it does not support a face editing workflow.

Other papers supplement decomposition approaches with the extraction of fine details, allowing to reconstruct a faithful facial model [6, 13, 22]. The major problem with these approaches is that they work for a specific person and do not provide editing capabilities. The Phace [14] method allows the user to edit fat or muscle maps in texture spaces on the face. While this provides a physically-based adjustment, the control is implicit. The user modifies the texture and then the system simulates muscles and fat to get the result.

Wu et al. [27] propose an anatomically-constrained local deformation model to improve the fidelity of monocular facial animation. Their model uses 1000 overlapping parts, and then decouples the rigid pose of the part from its non-rigid deformation. While this approach works particularly well for reconstruction, the parts are too small for editing semantic face parts such as the nose or the eyes.

Contrary to the methods described thus far, the Allen et al. [1] and BodyTalk [25] methods greatly facilitate editing by mapping intuitive features to modifications of global 3DMM eigenvector weights. In particular, BodyTalk [25] relates transformations of the meshes to keywords such as "fit" and "sturdy". While the mapping between the words and the deformations is not perfect, it still makes it reasonably intuitive to edit the mesh of the body. One problem with this method is that it provides words for bodies, not faces. A second major problem is the inability to make local adjustments, and adjustments that increase the length of the legs will result in changes to other regions such as the torso and arms. In contrast, for our approach, we aim at providing local control in the editing.

A downside of global PCA-based methods is that they exhibit global support: adjusting parameters to change one part has unwanted effects on other unrelated parts. To address this problem, we segment the face into independent sub-regions and provide a process to select the best set of eigenvectors, given a target number of eigenvectors. Methods that segment the face in sub-regions target facial animation instead of modeling. We will demonstrate that our approach is better-suited to the task of face editing than these methods. Another problem with most of the previous related works is that they do not allow facial model editing through the adjustment of objective measurements. In contrast, our method relies on anthropometric measurements used as controls for editing. Furthermore, we propose a process to select the right set of anthropometric measurements for each facial part.

## 3 OVERVIEW

In this paper, we introduce a pipeline for constructing a 3DMM. We separate the face into regions and compute independent PCA decomposition on each region. We then combine the per-region 3DMMs, paying particular attention to the selection of the most dominant eigenvectors across the eigenvectors of the different regions. While the eigenvectors are good at extracting the dominant data variation modes, they provide weak intuitive interpretation. We thus use anthropometric measurements to provide human understandable adjustments of the face. The reconstruction from the measurements is done through a mapping from the measurements to the weights that need to be applied to each eigenvector. From the set of measurements we extracted from our survey of the literature, we selected a subset which resulted in the least reconstruction error. An overview of our approach can be found in Fig. 1.

The remainder of this paper is organized as follows: Sec. 4 describes how 3DMMs are constructed, including face decomposition and selection of the most dominant eigenvectors. Afterwards, we discuss how to reconstruct a face by smooth blending of different facial parts (Sec. 5). In Sec. 6, the selection of the anthropometric measurements, and the mapping between these measurements and the PCA eigenvectors are discussed. We demonstrate the results in Sec. 7, and discuss them in Sec. 8.

## 4 3D MORPHABLE FACE MODEL

We employ PCA on a data set of faces to construct our 3DMMs. All faces are assumed to share a common mesh topology, with vertices in semantic correspondence. We propose to segment the face into different parts in order to focus the decomposition on a part-by-part basis instead of computing the PCA decomposition on the whole face. We compute the decomposition separately for the male and

female subsets. As shown in Fig. 1, we decompose the face into five parts: eyes, nose, mouth, ears, and what we refer to as the facial mask (which groups the remaining areas such as cheeks, jaws, forehead, and chin). We further discuss this design choice in Sec. 8.2. This face decomposition allows us to have eigenvectors for each part. The geometry of the facial parts is represented with a shape-vector $S_d = [V_1 \ldots V_{n_v}] \in R^{3n_v}$, where $n_v$ is the number of vertices of $d$th facial part, $d \in \{1, \ldots, 5\}$, and $V_i = [x_i y_i z_i] \in R^3$ defines the $x$, $y$, and $z$ coordinates of the $i$th vertex. After applying PCA, each facial part $d$ is reconstructed as:

$$S'_d = \overline{S_d} + \sum_{j=1}^{n_e} P_j b_j, \qquad (1)$$

where $\overline{S_d}$ is the mean shape of $d$th facial part, $n_e$ is its number of eigenvectors, $P_j$ is an eigenvector of size $3n_v$, $b$ is a $n_e \times 1$ vector containing the weights of the corresponding eigenvectors, and $S'_d$ is the reconstruction, which will be an approximation when not using all eigenvectors.

Our approach selects the smallest set of eigenvectors that still reconstructs the shape accurately. We accomplish this by incrementally adding the eigenvectors, in the order of their significance, to the reconstruction until a certain accuracy is met. Even though we rely on the eigenvalues to sort the eigenvectors for each part (largest to smallest eigenvalue), we provide the user with a measurable error (in mm), which is more precise than relying solely on eigenvalues across different parts. We determine the best set of eigenvectors to achieve a balance between the quality of the per-part reconstruction and the whole face reconstruction. To evaluate the accuracy of our selection, we construct the facial parts (Eq. 1) and blend them together (Sec. 5) to generate the whole face. Afterwards, we assess the accuracy of the reconstruction by calculating the average of the geometric error $D_{GE}$ between the ground truth and the blended face. We first do a rigid alignment step (rotation and translation) between the facial parts of the ground truth and the blended result. We then record the average per-vertex Euclidean distance over all vertices and per part:

$$D_{GE_{all}}(S') = \frac{1}{n_{all}} \sum_{d=1}^{5} \sum_{\substack{V'_j \in S'_d \\ V_j \in S_d}} \left\| V_j - \left( R_d V'_j + T_d \right) \right\| \qquad (2)$$

$$D_{GE_{part}}(S') = \frac{1}{5} \sum_{d=1}^{5} \frac{1}{n_d} \sum_{\substack{V'_j \in S'_d \\ V_j \in S_d}} \left\| V_j - \left( R_d V'_j + T_d \right) \right\| \qquad (3)$$

$$D_{GE}(S') = \frac{D_{GE_{all}}(S') + D_{GE_{part}}(S')}{2}, \qquad (4)$$

where $n_{all}$ is the number of vertices of the face mesh, $n_d$ is the number of vertices for part $d$, $V_j$ is on the ground truth and $V'_j$ is the corresponding point on the blended face. We compute averages over all vertices and per part to ensure that parts with more vertices do not end up using most of the eigenvector budget at the expense of parts with fewer vertices. We do so for the entire data set and for a set of 19 validation faces that were not part of the training data set. We compute the median data set error as well as the median validation error, and we average the two in a global error. The process of reconstructing the parts of validation faces is done by projecting each part onto the corresponding eigenvector basis (followed by the blending process).

At each step of our incremental eigenvector selection, we decide which of the five parts will get a new eigenvector added to its set. We compare the geometric errors resulting from each of the five candidate eigenvectors, and we select a candidate eigenvector

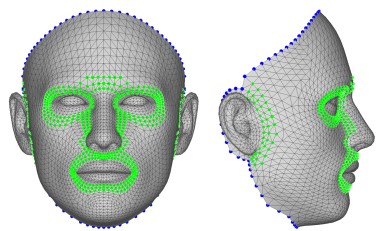

Figure 2: Regions highlighted in green and blue contain the transition and fixed zones, respectively.

which has a great impact on decreasing the error. When eigenvectors from multiple parts result in similar decreases in error, instead of systematically picking the eigenvector based on lowest error, we select by sampling from a discrete probability density function (PDF) created from the respective decreases in error of the five candidate eigenvectors. This PDF selection process creates a more even distribution of eigenvectors across the parts and maintains a low error. As we iterate, the reconstruction error decreases. For the female and male data set faces, the average reconstruction errors are 2.00 and 2.13 mm when considering zero eigenvectors. The errors decrease to 0.75 and 0.74 mm after 80 iterations, and when considering all eigenvectors the errors are 0 mm. We chose an error threshold of 1 mm which balances out the cost associated with considering too many eigenvectors and the accuracy of the reconstruction. Table 1 shows the resulting eigenvector distribution after achieving our 1 mm reconstruction accuracy. We experimented with reconstructing the female and male validation faces based on using our subset of eigenvectors. The median reconstruction errors are 1.33 and 1.48 mm, respectively.

Table 1: Number of eigenvectors selected for each part

| Facial part | # Eigenvectors for female | # Eigenvectors for male |
|---|---|---|
| Facial mask | 7 | 9 |
| Eye | 10 | 5 |
| Nose | 6 | 6 |
| Mouth | 9 | 10 |
| Ear | 14 | 16 |

## 5 FACE CONSTRUCTION THROUGH PARTS BLENDING

This step focuses on the problem of constructing a realistic new face by blending the five segmented parts together. As opposed to methods such as those of Tena et al. [26] and Cao et al. [7], which handle the transition between the parts by relying on the adjustment of a single strip of vertices, we spread the transition across three strips of vertices. In contrast to other methods that adjust the transition by vertex averaging [7] or least-squares fitting [26], we use Laplacian blending [24] of the parts and the transition, resulting in a smooth, yet faithful global surface. The vertex positions are solved by an energy minimization which reduces the surface curvature discontinuities at the junction between the parts while maintaining the desired surface curvature. To this end, we define a transition zone made of quadrilateral strips around the parts. In our experiments, a band of two quadrilaterals (three rings of green vertices in Fig. 2) provides good results. We interpolate the Laplacian $\mathscr{L}$ (the cotangent weights) of the five facial parts weighted by $\beta_d$, which has values of $\beta_d = 1$ inside the part, $\beta_d \in \{0.75, 0.5, 0.25\}$ going outward of the part in the transition zone, and $\beta_d = 0$ elsewhere. We normalize these weights such that they sum to one for each vertex. These soft

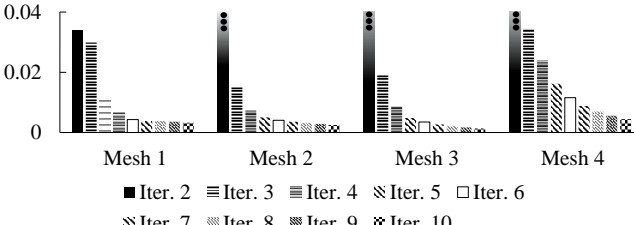

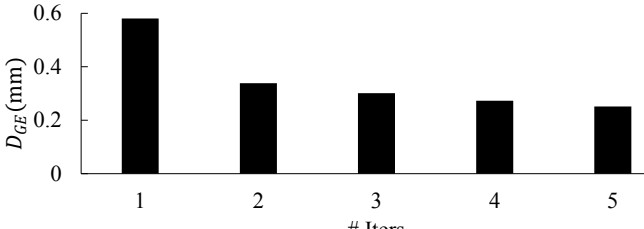

Figure 3: Graph showing the evolution of the Frobenius norm of the rotation between two consecutive iterations (averaged across the five rotations $R_d$). For each of the meshes 1 to 4, we begin with the average parts and change the weight of one eigenvector per part. Each eigenvector is selected randomly (from the first ten eigenvectors if there are more than ten eigenvectors for the part). The new value for the weight is also randomly selected within the range of $-2$ and $+2$ times the standard deviation for this eigenvector.

Figure 5: Graph comparing the average geometric error (in mm) between the ground truth parts and their blended counterparts, for different numbers of iterations.

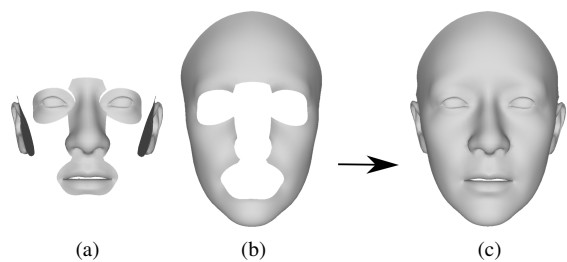

Figure 4: (a) and (b) are the generated parts. As in Fig. 3, we modified each average part by changing the weight of one eigenvector selected randomly. The new value for the weight is also randomly selected. (c) shows the result of blending the facial parts.

Table 2: Anatomical terms and corresponding abbreviations of our selected and discarded measurements.

**Selected measurements**

| Anatomical term | Abbrev | Ref |
| --- | --- | --- |
| Nasal Width÷Root Width | NWRW | [10] |
| Nasal Width÷Length of Bridge | NWLB | [10] |
| Nasal Width÷Width of Nostril | NWWN | [10] |
| Nasal Root Width÷Tip Protrusion | NRTP | [10] |
| Length of Nasal Bridge÷Tip Protrusion | NBTP | [10] |
| Nasal Width÷Tip Protrusion | NWTP | [10] |
| Nasal Root Width÷Length of Bridge | NRLB | [10] |
| Nasal Root Width÷Width of Nostril | NRWN | [10] |
| Length of Nasal Bridge÷Width of Nostril | NBWN | [10] |
| Width of Nose÷Tip Protrusion | WNTP | [10] |
| Philtrum Width | PW | [11] |
| Face Height | FH | [12] |
| Orbits Intercanthal Width | OIW | [12] |
| Orbits Fissure Length | OFL | [12] |
| Orbits Biocular Width | OBW | [12] |
| Nose Height | NH | [12] |
| Face Width | FW | [15] |
| Bitragion Width | BW | [15] |
| Ear Height | EH | [15] |
| Bigonial Breadth | B | [16] |
| Bizygomatic Breadth | BB | [16] |
| Facial Index | F | [21] |
| Nasal Index | N | [21] |
| Mouth-Face Width Index | MFW | [21] |
| Biocular Width-Total Face Height Index | BWFH | [21] |
| Lip Length | LL | [29] |
| Maximum Frontal Breadth | Max FB | [29] |
| Interpupillary Distance | ID | [29] |
| Nose Protrusion | NP | [29] |
| Nose Length | NL | [29] |
| Nose Breadth | NB | [29] |

**Discarded measurements**

| Anatomical term | Abbrev | Ref |
| --- | --- | --- |
| Eye Fissure Index | EF | [21] |
| Minimum Frontal Breadth | Min FB | [29] |

constraints allow some leeway in the transition zone. The boundary conditions of our system are set to the ring of blue vertices in Fig. 2, and we solve for the remaining vertices. To this end, we minimize the following energy function:

$$E(V') = \sum_{i \in \text{inner}} \left\| T_i \mathscr{L}(V_i') - \frac{\sum_{d=1}^{5} \beta_{i,d} R_d \mathscr{L}(V_{i,d})}{\sum_{b=1}^{5} \beta_{i,b}} \right\|^2, \quad (5)$$

where "inner" is the set of vertices of the five parts, excluding the vertices of the boundary conditions; $T_i$ is an appropriate transformation for vertex $V_i'$ based on the eventual new configuration of vertices $V_i$ and $R_d$ is the rotation of part $d$.

We solve Eq. 5 in a similar fashion to ARAP [23] by alternating solving for the vertex position and rotation matrices until the change is small. Fig. 3 shows that the rotation quickly converges as the Frobenius norm of consecutive rotations is large only for the first few iterations. Given our experiments, we decided to stop iterating when the Frobenius norm fell below 0.01 or after 6 iterations. Fig. 4 shows an example of a blended face. In this case, the Frobenius norm was below 0.01 after five iterations. Fig. 5, shows the evolution of the geometric error $D_{GE}$ between the ground truth parts and their blended counterparts for the example of Fig. 4. As can be seen, the error quickly reaches a plateau as the rotation stabilizes.

## 6 SYNTHESIZING FACES FROM ANTHROPOMETRIC MEASUREMENTS

PCA eigenvectors characterize the data variation space, but do not provide a clear intuitive interpretation. In this paper, we focus mainly on constructing linear regression models from data using a set of intuitive facial anthropometric measurements. Facial anthropometric measurements provide a quantitative description by means of measurements taken between specific surface landmarks defined with respect to anatomical features. We use the 33 parameters listed in Table 2. Each measurement corresponds to either a Euclidean distance or a ratio of Euclidean distances between surface positions, as specified in each paper cited in Table 2. In this section, we propose a measurement selection technique which assesses the accuracy

of each measurement, resulting in the most relevant ones for each facial part.

## 6.1 Mapping Method

We evaluate the measurements on the facial parts of the data set, yielding $f_{d,i} = \left[f_{i_1} \ldots f_{i_{n_m}}\right]$ for the $d$th facial part of scan $S_i$ considering $n_m$ measures. The measures for all of the scans are combined into an $n_m \times n_s$ matrix, $F_d = \left[f_{d,1}^T \ldots f_{d,n_s}^T\right]$, where $n_s$ is the number of scans. We learn how to adjust the weights of the PCA eigenvectors to reconstruct faces having specific characteristics corresponding to the measures. We adopt the general method of Allen et al. [1]. However, while that method learns a global mapping that adjusts the whole body, we will learn per-part local mappings. Furthermore, in Sec. 6.2 we will derive a process to select the best measures out of the set of all measures $\left[f_{i_1} \ldots f_{i_{n_m}}\right]$, and will proceed independently for each of the five parts.

We relate measures by learning a linear mapping to the PCA weights. With the $n_m$ measures for the $d$th facial part, the mapping will be represented as a $(n_e) \times (n_m + 1)$ matrix, $M_d$:

$$M_d \left[f_{i_1} \ldots f_{i_{n_m}} 1\right]^T = b, \tag{6}$$

where $b$ is the corresponding eigenvector weight vector. Collecting the measurements for the whole data set, the mapping matrix is solved as:

$$M_d = B_d F_d^+, \tag{7}$$

where $B_d$ is a $(n_e) \times (n_s + 1)$ matrix containing the corresponding eigenvector weights of the related facial part and $F_d^+$ is the pseudoinverse of $F_d$. As in Eq. 6, a row of 1s is appended to the measurement matrix $F_d$ for y-intercepts in the regression.

To construct a new facial part based on specific measurements, we use $b$ in Eq. 1, as follows:

$$S_d' = \overline{S_d} + Pb, \tag{8}$$

where $\overline{S_d}$ is the mean shape of the $d$th facial part and $P$ is the matrix containing the eigenvectors. Moreover, we can define delta-feature vectors of the form:

$$\Delta f_d = \left[\Delta f_1 \ldots \Delta f_{n_m} 0\right]^T, \tag{9}$$

where each $\Delta f$ contains the user-prescribed differences in measurement values. Afterwards, by adding $\Delta b = M_d \Delta f_d$ to the related eigenvector weights, it is possible to adjust the measure such as to make a face slimmer or fatter.

## 6.2 Measurement Selection

We propose a novel technique for automatically detecting the most effective and relevant anthropometric measurements. Some might be redundant with respect to others, some might not make sense for a specific part (e.g., the "Ear Height" might not be relevant for the mouth), and some might even lead to mapping matrices that generate worst results. During our investigations, we discovered that considering more anthropometric measurements does not necessarily lead to a lower reconstruction average error. Fig. 6 illustrates that a higher error occurs considering all the measurements in comparison with our selected combination. In order to aggregate the error for the given part, we reconstruct the face, relying only on the anthropometric measurements of the selected part, and then calculate the average error, as in Sec. 4. Fig. 7 shows two odd-looking examples from using all the measurements of the nose (a) and facial mask (b). We thus evaluate the set of relevant measurements, separately for each part. We begin with an empty set of *selected measurements*, and we iteratively test which measurement we should add to the set by evaluating the quality of the reconstructed faces when creating the mapping matrix, considering the currently selected measurements

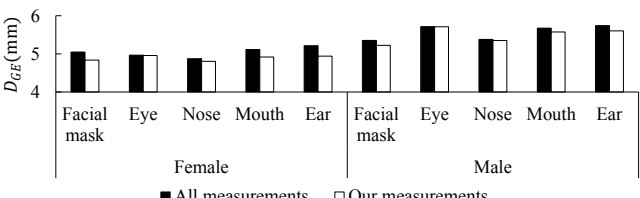

Figure 6: Using all measurements leads to higher reconstruction errors (mm) as compared to our set of selected measurements on the data set and validation faces

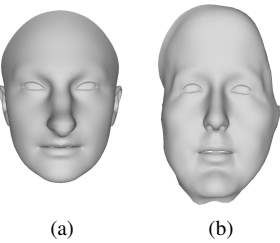

(a)        (b)

Figure 7: Using all the semantic measurements of the nose (a) and facial mask (b) often leads to odd-looking parts when editing through the adjustment of measurement values

together with the *candidate measurement*. We reconstruct a face using the mapping matrix (Eq. 6, 8) based only on its measurement values. The reconstructed face is considered as a *prediction*, and thus we evaluate the prediction quality in a fashion very similar to that used for eigenvector selection, by reconstructing all of the faces found in the data set of facial scans, as well as the 19 validation faces.

Each candidate measurement is used together with the current set of selected measurements, and we compute the candidate mapping matrix from this set of measurements. We use the mapping matrix with the data set and validation faces, and reconstruct all of the instances of the part under consideration (e.g., all of the mouths). We then evaluate a geometric error, $D_{GE}$, with the per-vertex distance between each predicted instance and its corresponding ground truth instance. The distance is calculated after a rigid alignment of the predicted instance to the ground truth instance is performed. We can thus ensure that we are evaluating the fidelity of the shape, and not its pose. If one or a few faces result in a large error, this could lead to the rejection of a measurement, which might still be beneficial for the prediction of most faces. To avoid this, we also measure the percentage $D_{NI}$ of faces for which an error improvement is seen. We count the number of faces whose geometric errors have been decreased by considering the candidate measurement. We then normalize $D_{GE}$ and $D_{NI}$ to the $[0,1]$ range and combine them into a single reconstruction quality measure:

$$quality = normalize(D_{NI}) + 1 - normalize(D_{GE}). \tag{10}$$

Considering the combined geometric error and percentage of improvement of all candidate measurements, we pick the one which will be added to the set of selected measurements. We stop adding measurements when we observe an increase of $D_{GE}$ and a value $D_{NI}$ below 50%. We repeat this process for each part (eyes, nose, mouth, etc.)

The selected anthropometric measurements are enumerated in Table 3. The description of each measurement, as well as the reference to the literature from which we obtained the measurement, are shown in Table 2, where we also list the measurements we rejected (measurements which were never selected for any of the segments).

Table 3: Combinations of anthropometric measurements

**For female**

| Part | Selected measures |
|------|-------------------|
| Facial mask | B, BB, BW, BWFH, FH, MaxFB, NBWN, NH, NP, PW, WNTP |
| Eye | B, BB, BW, BWFH, EH, F, FW, ID, LL, NB, NBTP, NH, NRLB, NRTP, NWRW, NWWN, OBW, OFL, OIW, PW |
| Nose | EH, LL, N, NB, NBTP, NBWN, NH, NL, NP, NWTP, NWWN, PW |
| Mouth | BWFH, F, LL, MFW, NP, NRWN, NWTP, PW |
| Ear | EH, FH, FW, MaxFB, NB, NBTP, NBWN, NP, NRLB, NRTP, NWLB, NWTP, OBW, PW, WNTP |

**For male**

| Part | Selected measures |
|------|-------------------|
| Facial mask | BW, BWFH, F, FH, FW, MaxFB, NRWN, OBW, OFL, PW |
| Eye | B, BB, BW, F, FW, ID, N, NBTP, NBWN, OBW, OFL, PW |
| Nose | BB, FW, ID, LL, MaxFB, NB, NBWN, NH, NP, NRLB, NRTP, NWLB, NWWN, OBW, OIW |
| Mouth | B, BW, BWFH, F, LL, NBTP, NH, PW |
| Ear | B, BB, BWFH, EH, FH, MaxFB, N, NRTP, NWRW, OBW, OIW |

## 6.3 Correlation Between Measurements

Defining the correlation between the measurements is important for the adjustment of faces. Accordingly, if the user adjusts one measurement, the system automatically calculates the adjustment of the other measurements as well. This greatly helps to create realistic faces by maintaining the correlation observed in the data set. Similarly to Body Talk [25], we use Pearson's correlation coefficient on $F$ to evaluate the relationship between the anthropometric measurements. Considering a facial part $d$, the Pearson's correlation coefficient $\text{Cor}_{jk}$ for measurements $j$ and $k$ is expressed as:

$$\text{Cor}_{jk} = \frac{\sum_{i=1}^{n_s}(f_{ij} - \overline{f_j})(f_{ik} - \overline{f_k})}{\sqrt{\sum_{i=1}^{n_s}(f_{ij} - \overline{f_j})^2}\sqrt{\sum_{i=1}^{n_s}(f_{ik} - \overline{f_k})^2}}, \qquad (11)$$

where $f_{ij}, f_{ik} \in f_{d,i}$ are measurements of scan $S_i$, $\overline{f_j}$ and $\overline{f_k}$ are the mean values of measurements $j$ and $k$ respectively, and $n_s$ is the number of scans. The coefficient is a value between $-1$ and $1$ that represents the correlation. When adjusting measurement $k$ by $\Delta f_k$, we get the change in other measures as $\Delta f_j = \text{Cor}_{jk}\Delta f_k$. Accordingly, we can evaluate the influence of one measurement on the others, as well as the conditioning on one or more measurements, and create the most likely ratings of the other measurements.

## 7 RESULTS

Compared to global 3DMM methods that compute one set of eigenvectors for the whole face, our 3DMM computes a set of eigenvectors for each part. This is at the root of one of the advantages of our approach: its ability to locally adjust faces. We compare our approach to other methods that rely on local 3DMM. We created mapping matrices (Eq. 7) for global 3DMMs, SPLOCS [19], clustered PCA [26], as well as our part-based 3DMMs, and tested the adjustment of measurements with these models. We used 46 eigenvectors for global 3DMM, SPLOCS, and our part-based 3DMM. For clustered PCA, we first tested using 13 clusters, as is reported in the paper, but found that this leads to a non-symmetrical result (Fig. 8b). By checking other clusterings, we selected 12 clusters (Fig. 8a). Because a clustered PCA does not allow for a different

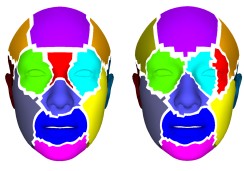

(a) 12 clusters (b) 13 clusters

Figure 8: Automatic part identification of clustered PCA [26]. Note how the automatic clustering leads to non-symmetrical clusters (left eye with one cluster vs. right eye with two clusters) for 13 clusters and required us to manually check which other clustering would be usable.

number of eigenvectors for each cluster, and to avoid having too few eigenvectors per part, we used 46 eigenvectors for each cluster (selecting the 46 with the largest eigenvalues).

To compare our approach and the use of measurements with other methods, we decided on a way to use our measurements with SPLOCS and clustered PCA. We further demonstrate that with SPLOCS and our approach, we can have more *local measurement* or *global measurement* control. For our approach, Table 3 shows that some measures influence more than one part. For example, the "Lip Length" is found in the lists for both mouth and nose. When a measurement is shared between different facial parts, our method allows to decide to have more localized changes by adjusting the measure for only one part, or to have more coherence across the parts by adjusting all of the parts involved in the measurement. If comparing with SPLOCS, we can also balance between local measurements and global measurements. Each measurement is based on computations involving specific *measurement vertices* (such as the corner of the mouth and the tip of the nose). To enforce locality, when considering a measurement, we check which SPLOCS "eigenvectors" infer significant movement at the related *measurement vertices*. We compute this by checking if the eigenvector displacement vector at a *measurement vertex* is large enough as compared to the maximum displacement vector of the eigenvector (we check if it is larger than 1% of the maximum displacement of all vertices of the eigenvector). A SPLOCS eigenvector is considered for a measurement only if it meets the criterion for one of the *measurement vertices* of a specific measurement. To enforce more globality with SPLOCS, we use the mapping matrices for all of the eigenvectors. Fig. 7 shows an example of the globality and locality of the influence of adjusting the "Lip Length". It compares global PCA eigenvectors, local measurement and global measurement SPLOCS, clustered PCA, and our local measurement and global measurement approaches. The color coding shows the per-vertex Euclidean distance. Note that the colors do not represent errors, but rather, vertex movements. Thus, the goal is to have warmer colors around the location where the editing is intended, and colder colors in unrelated regions. Our method allows having global measurement influenced by adjusting the measure for both the nose and the mouth parts, as well as more localized changes by adjusting only the mouth (Fig. 9c-9f). Contrary to our approach, both global measurement and local measurement SPLOCS resulted in similar deformations all over the face, while the expected result was a modification focused around the mouth (Fig. 9b-9d).

We will now focus on local measurement editing. Fig. 10-12 show the adjustment of the same anthropometric measurement using global 3DMM, local measurement SPLOCS, clustered PCA, and our local measurement approach. In Fig. 10f-10g, we can see that even though we wanted to adjust the "Nose Breadth", the adjustment using the global eigenvectors and local measurement SPLOCS resulted in significant deformations all over the face, while clustered PCA and our approach could focus the deformation around the nose, as expected (Fig. 10h-10i). We can observe similar un-

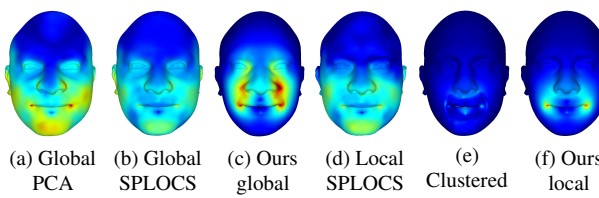

(a) Global PCA | (b) Global SPLOCS | (c) Ours global | (d) Local SPLOCS | (e) Clustered | (f) Ours local

Figure 9: Comparison of the globality vs. locality of the adjustments (editing by increasing the "Lip Length"): (a) global PCA eigenvectors, (b) global measurement SPLOCS, (c) our global measurement approach, (d) local measurement SPLOCS, (e) clustered PCA, and (f) our local measurement approach. The colors respectively represent per-vertex Euclidean distance (blue = 0 mm, red = 8.5 mm). Note how our local measurement and global measurement approaches induce significant and local surface deformation to achieve the desired editing. In comparison, global PCA and SPLOCS induce non-local deformation, and clustered PCA induces much less deformation.

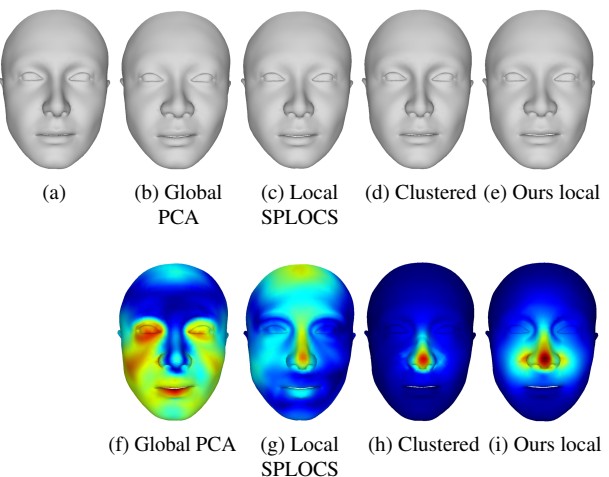

(a) | (b) Global PCA | (c) Local SPLOCS | (d) Clustered | (e) Ours local

(f) Global PCA | (g) Local SPLOCS | (h) Clustered | (i) Ours local

Figure 10: "Nose Breadth" adjustment results: (a) nose of a female from validation faces adjusted using (b) global PCA eigenvectors, (c) SPLOCS, (d) clustered PCA, and (e) our approach. The color mapped renderings (f)-(i) indicate respective per-vertex Euclidean distance (blue = 0 mm, red = 5 mm).

wanted global deformations of the face in Fig. 11f-11g. Also note that the automatic segmentation of clustered PCA does not provide the desired deformation for some cases, such as in Fig. 11h-12h. We consistently outperform clustered PCA in terms of local deformation where expected. The results shown in Fig. 10-12 highlight the difficulty of locally controlling the face deformation, and the power of our approach in locally adjusting the face with respect to the anthropometric measurements.

In the accompanying video, we show multiple edits on multiple parts, starting from the average face, while Fig. 13 shows edits starting from four real faces. We can see that our approach allows capturing the essence of the anthropometric measurements, providing an easy-to-use workflow.

## 8 DISCUSSION

In this section, we discuss different aspects of our approach. We present different comparisons highlighting the impact of the eigenvector and measurement selection. We then discuss the face segmentation choice, and end by describing the procedure used to bring all of our scans to a common face mesh.

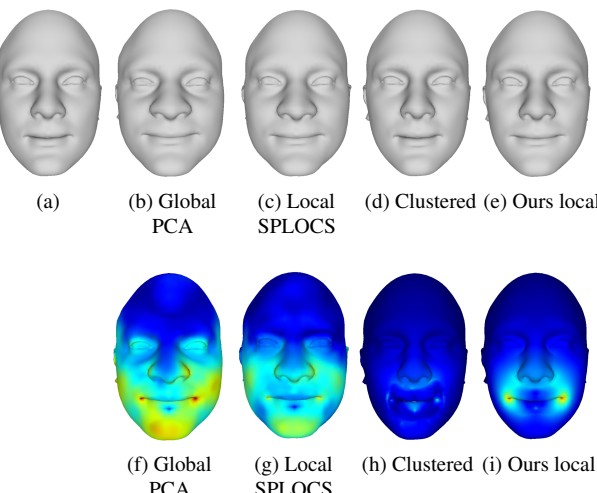

(a) | (b) Global PCA | (c) Local SPLOCS | (d) Clustered | (e) Ours local

(f) Global PCA | (g) Local SPLOCS | (h) Clustered | (i) Ours local

Figure 11: "Lip Length" increase results: (a) mouth of a male from validation faces edited using (b) global PCA eigenvectors, (c) SPLOCS, (d) clustered PCA, and (e) our approach. The color mapped renderings (f)-(i) indicate respective per-vertex Euclidean distance (blue = 0 mm, red = 8 mm).

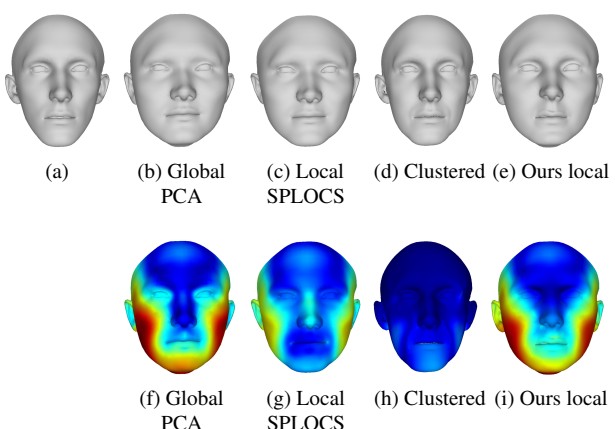

(a) | (b) Global PCA | (c) Local SPLOCS | (d) Clustered | (e) Ours local

(f) Global PCA | (g) Local SPLOCS | (h) Clustered | (i) Ours local

Figure 12: "Bizygomatic Breadth" (the bizygomatic width of the face) increase results: (a) a male from the validation faces edited using (b) global PCA eigenvectors, (c) SPLOCS, (d) clustered PCA, and (e) our approach. The color mapped renderings (f)-(i) indicate respective per-vertex Euclidean distance (blue = 0 mm, red = 14 mm).

### 8.1 Measurements Error

To verify the robustness of our 3DMMs and of our set of selected measurements, we reconstruct real faces, relying on their anthropometric measurements to compute their eigenvector weights (Eq. 6). We then get the face with our approach, including the blending procedure (Sec. 5), and compute its resulting anthropometric measurements. We compute the quality of the reconstruction through the absolute value of the difference between the ground truth measurement and the measurement from the reconstructed face. Since measurements correspond either to a Euclidean distance or to a ratio of Euclidean distances, we normalized all the measurements to the $[0\%, 100\%]$ range. Fig. 14 shows that the average percentage of error is low when using "our measurements". This means that both the selection of eigenvectors and the mapping matrix work well. Furthermore, it shows that when using "all measurements" to

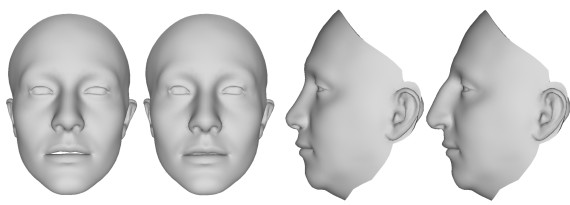

(a) -"Orbits Biocular Width" of eye   (b) +"Nose Protrusion" of nose

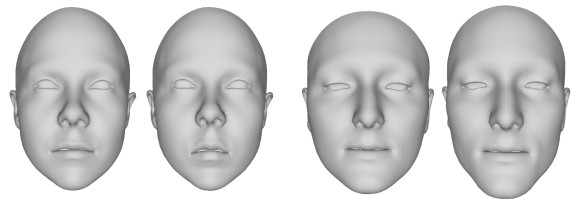

(c) -"Philtrum Width" of mouth   (d) +"Face Height" of facial mask

Figure 13: We generated random faces (left faces (a)-(d)) and edited them by increasing ("+") or decreasing ("-") the value of some of the indicated anthropometric measurements

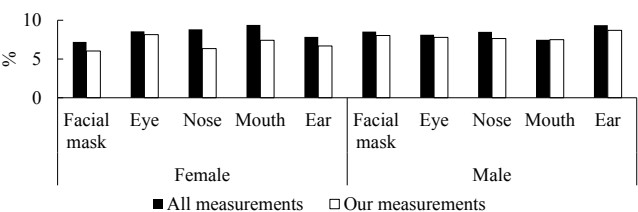

Figure 14: Using our subset of measurements on the data set and validation faces leads to lower errors (percentage), as compared to using "all measurements"

compute the mapping matrix (Eq. 7), we get larger average errors as compared to ground truth measurements. When calculating the error in Fig. 14 for "our measurements", we calculate the average error over our selected measurements only (Table 3). The error shown in Fig. 14 for "all measurements" also considers only our selected measurements (if the error across all of the measurements is considered, the comparison is even more in favor of using our selected measurements).

We evaluated how our approach compared to SPLOCS and clustered PCA with respect to achieving measurement values prescribed by edit operations. We created a set of 1,000 random edits on 135 face meshes. We took the resulting edited face mesh from our approach, SPLOCS, as well as clustered PCA, and evaluate the difference between the measurement value prescribed by the editing and the measurement value calculated from the edited mesh. Overall, our approach is the one that performed the best, with the resulting measurement being closest to the prescribed measurement. SPLOCS was second and clustered PCA presented the greatest differences (see Fig. 15).

Even though our approach is the one that is closest (on average) to the prescribed measurements, there is a limitation due to the blending of the synthesized parts. This blending sometimes affects the mesh in a way that prevents it from achieving the exact prescribed effect for the editing. Fig. 16 shows an example where the blending does not maintain the "Nose Height" of the synthesized nose as it deforms it through the blending process.

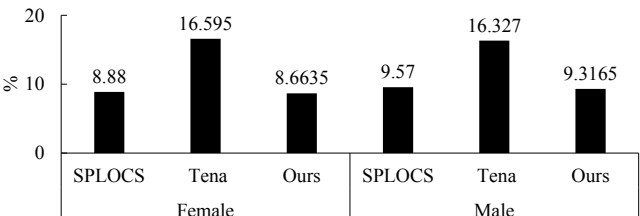

Figure 15: Starting from one face, we adjust one of its measurements to match the value of that measurement for another face. We then compute the difference between the prescribed measurement value and the measurement value calculated from the mesh. We do so for 1,000 such edits. Our approach leads to a smaller error (percentage), as compared to the clustered PCA, and to slightly better results when compared to local measurement SPLOCS.

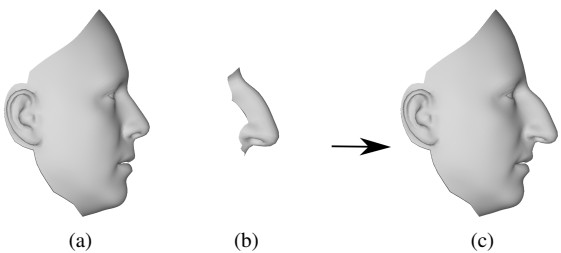

(a)                    (b)                    (c)

Figure 16: (a) Average male head. Its "Nose Height" is 45.09 mm. (b) A synthesized nose with its "Nose Height" edited to 70.11 mm. (c) Result of blending the nose. While this is an extreme case, it still reflects the fact that the approach is not always able to achieve the prescribed measurement (the value decreased to 58.53 mm for this example).

## 8.2   Face Decomposition

Our face segmentation was motivated by several facial animation artists with whom we worked, and who strongly prefer having control over the face patches in order to make sure they match the morphology of the face and muscle locations. This type of control is impossible to achieve with an automatic method, which is typically agnostic to the underlying anatomical structure. It is important to note that this manual way of selecting the regions is no more cumbersome than the current state-of-the-art methods. The state-of-the-art method of Tena et al. [26] requires a post-processing step to fix occasional artifacts in the segmentation method. Furthermore, as illustrated in Fig. 8b, segmentation boundaries can occasionally occur across important semantic regions such as the eyes, leading to complications further down the pipeline.

## 8.3   Data Set

The quality of the input mesh data set is crucial for the reconstruction of good 3D face models. As do many existing methods, we assume that the meshes share a common mesh topology. Mapping raw 3D scans to a common base mesh is typically done using a surface mapping method [2, 20, 28]. We established this correspondence with the commercial solution, R3DS WRAP. We plan to release a subset of our data set for other researchers.

## 9   CONCLUSION

In this paper, we designed a new local 3DMM used for face editing. We demonstrated the difficulty of locally editing the face with global 3DMMs; we thus segmented the face into five parts and combined the 3DMMs for each part into a single 3DMM by selecting the best

eigenvectors through prediction error measurements. We then proposed the use of established anthropometric measurements as a basis for face editing. We mapped the anthropometric measurements to the 3DMM through a mapping matrix. We proposed a process to select the best set of anthropometric measurements, leading to improved reconstruction accuracy and the removal of conflicting measurements. From a list of 33 anthropometric measurements we surveyed from the literature, we identified 31 which lead to an improvement of the reconstruction and rejected 2 as they decreased the quality of the reconstruction. Note that the anthropometric measurement selection process would apply as well even if using a different 3DMM from the one proposed in this paper, as well as when considering a different set of anthropometric measurements. We demonstrated this by applying our set of measurements to both SPLOCS [19] and clustered PCA [26]. This also demonstrated that our approach produces results superior to those of established methods proposing automatic segmentation and different ways to construct the eigenvector basis. We also presented different bits of experimental evidence to demonstrate the superiority of our approach, especially in terms of local control, as compared to the typical global 3DMM.

A limitation of our approach lies in the mapping matrices, which assume a linear relationship between anthropometric measurements and the eigenvector weights. An interesting avenue for future work would be to apply machine learning to identify non-linear mappings. Also, our measurements are based on distances between points on the surface. Future work could consider measurements based on the curvature over the face, such as measurements specifying the angle formed at the tip of the chin.

Although anthropometric measurements generate plausible facial geometric variations, they do not consider fine-scale or coarse-scale features. Regarding the fine-scale details, our approach does not model realistic variations of wrinkles, and that could be an interesting direction for future research. Regarding coarse-scale features, we could reconstruct a skull based on the anthropometric measurements, and then generate the facial mask based on an energy minimization of the skin thickness considering the skull and the measurements.

### ACKNOWLEDGMENTS

This work was supported by Ubisoft Inc., the Mitacs Accelerate Program, and École de technologie supérieure. We would like to thank the anonymous reviewers for their valuable comments.

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
