# OpenReview forum: "Part-Based 3D Face Morphable Model with Anthropometric Local Control"
_graphicsinterface.org/Graphics_Interface/2020/Conference — GI 2020_

### Official Review · AnonReviewer2 · 2020-01-09

**Confidence:** 3
**Rating:** 6

**Review:**

The paper proposes a method for improved editing of a morphable 3D face model (3DMM), leveraging a part-based decomposition of the face mesh. Edits are performed through a set of anthropometric measurement parameters selected  using an iterative geometric reconstruction error-guided selection process. These selected anthropometric measurements are then mapped onto a set of per-part Eigenvectors which were obtained through per-part PCA.

The proposed scheme improves the locality of edits and reduces the redundancy between different edit parameters, compared to prior work which typically employs method with global support.  The method is evaluated using a dataset of 135 training face scans and 19 validation face scans.  Qualitative results compare the proposed method with adapted versions of global SPLOCS (Neumann et al. 2013) and clustered PCA (Tena et al. 2011), demonstrating the improved locality of the approach.  Quantitate results show that the set of selected anthropometric measurements leads to lower overall reconstruction error on the validation faces, compared to using the full set of measurements as parameters, and that edits using the proposed method have lower error on validation faces compared to SPLOCS and clustered PCA.

The paper is relatively clear, with a few minor issues that could be fixed to improve clarity (see list below).  Though I am not a domain expert, I believe the work presents an original method.  On the positive side, this is a simple, seemingly technically sound approach that addresses practical problems with existing methods for editing morphable face models (namely locality of edits, and interpretable low-dimension edit parameters).  On the negative side, the presented method is quite straightforward, and the evaluation is carried out at a fairly shallow level.  It would have been nice to see the effectiveness of the proposed method be evaluated in practical use through a user study as well.

Given the above, I am borderline to slightly positive with respect to acceptance, as I believe the paper would be interesting to the community.


Minor issues with exposition
p2
and they control mechanism -> and the control mechanism
perform worst than -> perform worse than
to constructs realistic 3DMMs -> to construct realistic 3DMMs
decouple the rigid pose -> decouples the rigid pose
p3
geometry of the parts is presented -> geometry of the parts is represented
p4
Table 1: "46 steps" is unclear, should likely be explained in caption
p5
Figure 5: Add labels for the axes, and/or explain in caption
p6
Figure 6: Label for y axis. Also, it is unclear exactly how the given reconstruction errors are aggregated for the given parts.  Good to explain in caption/text
p7
Figure 9: The caption could be improved by stating a clear conclusion to be drawn from the comparisons
p9
leading to an improve reconstruction -> leading to improved reconstruction
p10
The last paragraph contains a couple of sentences that are disjointed and need to be rewritten: "... different parts that." and "Using a Generative Adversarial Network..."

---

### Official Review · AnonReviewer3 · 2020-01-09
**This work proposes a practically useful solution.**

**Confidence:** 3
**Rating:** 8

**Review:**

This paper describes a method of part-based morphable facial model allowing for localized user control. The method first splits a 3D face into pre-defined semantic parts. Then a PCA-based morphable model is constructed for each part. The best subset of anthropometric measurements is also selected, forming a mapping matrix. During the online stage, the user edits the facial model by prescribing the anthropometric measure values. The parts are then reconstructed with the mapping matrices. Finally, parts are fused together to form the final face model.

At first sight, the method design is ad-hoc. It relies on a predefined partition of facial models and involves a final step of part stitching. However, the fact that the existing models with localized control cannot handle well variations in terms of identity makes this paper well motivated and the solution respectful. The method is, albeit heuristic, practically useful and it demonstrates good results. The evaluation is quite extensive and satisfying. Overall, I am happy to see it published at GI.

I wonder if the part-based model can support realistic wrinkling. The authors should consider adding some discussion on this point. For the two limitations pointed out in the paper, some failure examples should be provided. In addition, I'd like to see to what degree the localized edit (in terms of the anthropometric measurements) can be supported by the model in order to avoid issues of part fusing.

---

### Official Review · AnonReviewer1 · 2020-01-09
**Good paper, recommend acceptance**

**Confidence:** 4
**Rating:** 8

**Review:**

This is a well-written paper that makes a meaningful contribution to the face modeling research area. It proposes a piecewise morphable model for human face meshes, with well-justified and reasonable methods for initial (manual) decomposition into "semantic", artist-friendly pieces, eigenvector basis selection for each piece, fitting to a target shape (but see comment on explaining how this is done, below), and blending independently reconstructed pieces together. It also proposes a mapping between anthropometric measurements of the face (e.g. lip length, nose width...) and the parameters of the model (building upon [ACP03] but with part-specific measurement saliency), in order to synthesize and edit faces with desired attributes. The authors evaluate many aspects of their method and compare to a good set of baselines.

I am not an expert in the specific area of face modeling, which has a pretty extensive history and lots of active work. With that caveat, I think this paper makes a solid contribution and I support acceptance.

- they control mechanism --> the control mechanism
- "Furthermore, we compared our approach": At this point the existence of "our approach" has not even been mentioned. Maybe move the rest of this para after the next one?
- Section 4 needs to clearly state right at the beginning that all faces in the dataset are assumed to have the same mesh topology, with vertices in semantic correspondence. Otherwise the rest of the section (e.g. the average per-vertex distance) is not justified.
- "we rely on the eigenvalues to sort the eigenvectors for each part": How is it sorted? The ordering of the eigenvectors from P_1 to P_{n_e} is not explained.
- "19 validation faces that were not part of the training data set": The process of fitting the model to a novel (non-training) face (presumably by projecting each part onto the current eigenvector basis followed by blending) is not actually described anywhere I think?
- zero eigenvector --> zero eigenvectors
- In Table 1 the column headers should probably be something like "Facial part", "#EVs for females", "#EVs for males"
- "Face Generation through Parts Blending" suggests a probabilistic generative model (as in statistics/ML) is developed from which novel faces are sampled. Since this is not what is described here, I would suggest renaming this section to "Blending Synthesized Parts into a Complete Face" or something of the sort.
- "Fig. 4 shows an example of blended face": Missing "a".
- "doing independently for each of the d parts" --> "doing this independently for each of the 5 parts" (d is the index of part, not the part count)
- At the end of Sec 5, I suggest showing some straight reconstruction examples on test faces. I.e. after selecting the final sets of per-part eigenvectors, try reconstructing a novel face mesh and measure the reconstruction error. This is a standard experiment in prior 3DMM papers, so might be useful to examine the fitting quality of the model ignoring the whole anthropometric measurement part.
- "6. Significance of Anthropometric Measurements": This is the title of an entire section, so should cover the whole content of that section. I'd suggest "Synthesizing Faces from Anthropometric Measurements" or something of the sort.
- Eq 8: Presumably P is a matrix stacking all eigenvectors, but this not actually defined anywhere I think.
- The math in Sec 6.1 is identical to [ACP03]. While this paper is cited here (so I don't accuse the authors of any plagiarism), the fact that the method is the same should be stated clearly. Maybe rephrase the intro sentence as: "We adopt the general approach of Allen et al. [ACP03]. However, while that method learns a global mapping that adjusts the whole body, we will learn per part local mappings. Furthermore..."
- "B_d is a (n_e ) × (n_s + 1) matrix containing the corresponding eigenvector of the related facial part": Do you mean containing the corresponding eigenvector weights? I think it would be clearer to first define the regression problem for each part in each shape: M_d [ f_1...f_{n_m} 1]^T = b. This is what is done in [ACP03].
- I may have missed something but what is "our global approach" in Fig 9?
- In Fig 15, does it make sense to also check that the other measurements _don't_ change? Since measurements are correlated, I'm not sure if this is a good or bad thing. Maybe the authors can comment.

---

### Meta-Review · Area_Chair1 · 2020-01-11

**Recommendation:** Accept
**Confidence:** 4

**Metareview:**

All 3 reviewers were positive about the paper, with two clear accepts and one weak accept. Hence, I recommend that the paper be accepted to GI, and request the authors to address all the issues identified by the reviewers in their final revision.

---

### Decision · Program_Chairs · 2020-01-11

Accept